# Clear and transparent nanocrystals for infrared-responsive carrier transfer

Masanori Sakamoto [1], Tokuhisa Kawawaki [1], Masato Kimura[2], Taizo Yoshinaga[3], Junie Jhon M. Vequizo [4], Hironori Matsunaga[4], Chandana Sampath Kumara Ranasinghe [4], Akira Yamakata [4], Hiroyuki Matsuzaki [5], Akihiro Furube[6] & Toshiharu Teranishi [1]

Infrared-light-induced carrier transfer is a key technology for 'invisible' optical devices for information communication systems and energy devices. However, clear and colourless photo-induced carrier transfer has not yet been demonstrated in the field of photochemistry, to the best of our knowledge. Here, we resolve this problem by employing short-wavelength-infrared (1400–4000 nm) localized surface plasmon resonance-induced electron injection from indium tin oxide nanocrystals to transparent metal oxides. The time-resolved infrared measurements visualize the dynamics of the carrier in this invisible system. Selective excitation of localized surface plasmon resonances causes hot electron injection with high efficiency (33%) and long-lived charge separation (~ 2–200 μs). We anticipate our study not only provides a breakthrough for plasmonic carrier transfer systems but may also stimulate the invention of state-of-the-art invisible optical devices.

[1] Institute for Chemical Research, Kyoto University, Gokasho, Uji, Kyoto 611-0011, Japan. [2] Department of Chemistry, Graduate School of Science, Kyoto University, Gokasho, Uji, Kyoto 611-0011, Japan. [3] Graduate School of Pure and Applied Sciences, University of Tsukuba, 1-1-1 Tennodai, Tsukuba, Japan. [4] Graduate School of Engineering, Toyota Technological Institute, 2-12-1 Hisakata, Tempaku, Nagoya 468-8511, Japan. [5] National Institute of Advanced Industrial Science and Technology (AIST), Tsukuba Central 2, 1-1-1 Umezono, Tsukuba, Ibaraki 305-8568, Japan. [6] Department of Optical Science, Tokushima University, 2-1, Minamijosanjima-cho, Tokushima 770-8506, Japan. Correspondence and requests for materials should be addressed to M.S. (email: sakamoto@scl.kyoto-u.ac.jp) or to T.T. (email: teranisi@scl.kyoto-u.ac.jp)

'I'nvisibility' has emerged as an important feature of photo-responsive materials, with their increasing demand in energy devices[1,2]. The control of light-absorption-induced carrier transfer is the bedrock of this subject. For the fabrication of invisible materials, an effective strategy is selective absorption of the ultraviolet (UV) or infrared (IR) region of light. As UV light is unfavourable for light to energy-conversion systems, IR-light absorbers are the key to responding to this challenge. However, simultaneous pursuit of clear and colourless IR-induced carrier transfer has been an important goal of photochemistry research.

The development of an IR-responsive pigment remains a great challenge. In terms of artificial materials, IR-responsive narrow band-gap semiconductors (e.g. InSb, HgCdTe, etc.) are opaque and exhibit dark colours derived from inter- and intra-band transitions.

Plasmonic materials, which are artificial pigments, exhibit optical properties overwhelmingly superior to those of natural pigments in the IR region[3–8]. The localized surface plasmon resonance (LSPR) band derived from the collective oscillation of carriers in transparent conductive-oxide nanocrystals (NCs) makes it possible to achieve selective absorption of short-wavelength infrared (SWIR) (1400–4000 nm) light, which is an important wavelength band in sensors[7–9].

Here we demonstrate SWIR-induced electron transfer from transparent indium tin oxide (ITO) NCs to metal oxides (SnO$_2$ and TiO$_2$). Time-resolved-IR spectroscopy of the ITO/SnO$_2$ heterointerface reveals high electron-injection efficiency (33%) and long-lived charge separation (~2–200 μs). Furthermore, we demonstrate the potential expansion of applicable IR light beyond 4 μm by using the tunability of LSPR of ITO NCs. We anticipate our result could constitute a step forward, not only in the science of plasmonic-carrier transfer, but also for state-of-the-art invisible optical devices in general.

## Results

**Fabrication and characterization of heterointerfaces.** For carrier injection using LSPR in the SWIR-region, the fabrication of heterointerfaces with rational-band alignment is essential. As electron-acceptor phases to form heterointerfaces with ITO NCs, we selected two types of metal-oxide semiconductors, TiO$_2$ (anatase or P25, a mixture of anatase and rutile) and SnO$_2$, because these metal oxides can be "clear and colourless" and are commonly used as electron-transport layers and/or photo-catalysts and possess suitable acceptor levels (i.e., position of the conduction band (CB)) for hot-electron injection from ITO NCs. Energy diagrams of the ITO NCs and metal oxides are shown in Fig. 1a. The energy difference (ΔE) between the Fermi level ($E_F$) of the ITO and the CBs of SnO$_2$ and TiO$_2$ are 0.2 and 0.7 eV, respectively, which are accessible for hot electrons generated in ITO NCs[10–12]. The combination of ITO and SiO$_2$ was also adopted as a monitoring reference for the LSPR-stimulated response of ITO NCs, because no electron transfer from the ITO NC to the insulating SiO$_2$ phase is expected. The ITO NCs were synthesised according to previous reports[13] and immobilised on metal oxides via thermal-annealing and a reductive-annealing process (see Methods). Figure 1b, c shows the UV–Vis–IR spectra of ITO NCs and three types of ITO/metal oxides. All samples have LSPR peaks in the IR region. TEM and XRD measurements revealed no changes in the size or crystalline structure of ITO NCs during the annealing process (Supplementary Figs. 1–5). The broadening of LSPR band of ITO NCs on SnO$_2$ results from the plasmonic coupling of ITO NCs (see Supplementary Fig. 5).

**Carrier dynamics at the heterointerfaces.** To elucidate the LSPR-induced carrier dynamics of ITO/metal oxides, we performed femtosecond (fs)- and microsecond (μs)-laser flash photolysis to obtain time-resolved infrared-absorption (TR-IR). TR-IR spectroscopy allows us to directly observe the LSPR-induced carrier dynamics at the ITO/metal-oxide heterointerfaces. The observation of free-carrier absorption (FCA) by metal oxides is

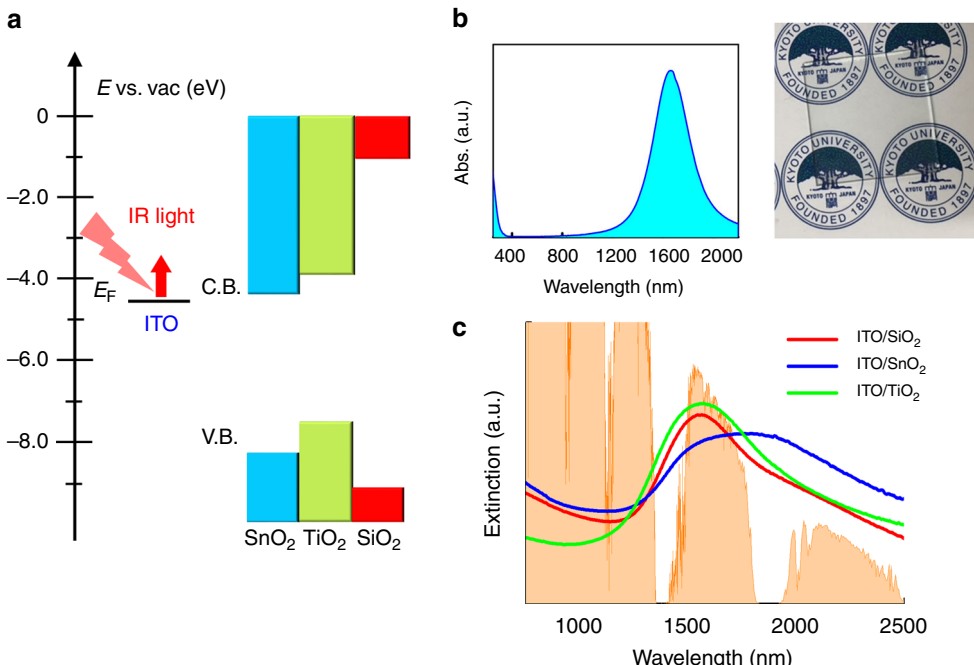

**Fig. 1** Energy diagrams and optical properties. **a** Energy diagrams of the indium tin oxide (ITO) nanocrystals (NCs) and metal oxides. The $E_F$ values of ITO NCs and the conduction band edges of SnO$_2$ and TiO$_2$ were obtained from references[10–12], respectively. C.B. conduction band; V.B. valence band. **b** Left-hand side image: absorption spectrum of ITO NCs in CHCl$_3$ solution. Right-hand side image: ITO-NC-coated glass substrate. **c** Extinction spectra of the ITO/SiO$_2$, ITO/TiO$_2$ and ITO/SnO$_2$ after reductive annealing. The spectrum shown in orange is the solar spectrum (AM 1.5)

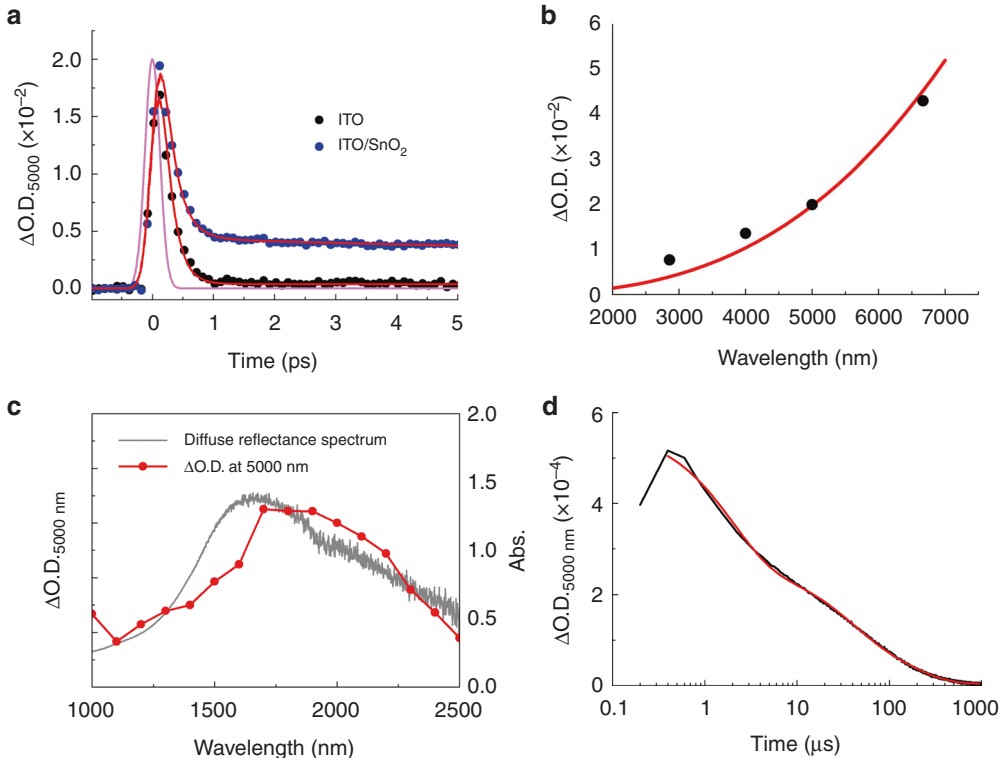

**Fig. 2** TR-IR measurements and excitation spectra. **a** Kinetic profiles for the transient absorption of ITO NCs and ITO/SnO$_2$ at the ps scale at 5000 nm upon the excitation by a 1700-nm laser. The instrument response function (IRF) (FWHM of IRF = 285 ± 40 fs) is shown by the pink line (see also Supplementary Figure 6). **b** TR-IR spectrum of ITO/SnO$_2$ at 10 ps after excitation of the LSPR band by a 1700-nm laser. The features of the observed FCA spectrum (black circles) agrees well with the absorption spectrum reproduced by simulation (red line)[15]. **c** Excitation spectrum for the FCA of SnO$_2$. The probe light had a fixed wavelength of 5000 nm and the excitation-light wavelength changed with the laser power kept at 6 μJ/pulse. **d** The kinetic profile of ITO/SnO$_2$ at 5000 nm in the μs region following excitation by the 1400-nm laser. (FWHM of IRF = 0.485 μs). The red line shows the line of best fit

the direct evidence of electron injection from the ITO NCs into the CBs of metal oxides.

The kinetic profiles of ITO/metal-oxide interfaces on a picosecond (ps) scale were measured to investigate light-stimulated electron injection from ITO to the metal oxide (Fig. 2a and Supplementary Fig. 7). Upon excitation of ITO and ITO/ SiO$_2$, transient absorptions (ΔOD) appear within the time resolution of 285 ± 40 fs and decay with the time constant of 0.17 ± 0.08 ps and 0.37 ± 0.11 ps, respectively (Supplementary Tables 1 and 2). The sequence of events in plasmonic materials following pump excitation includes electron dephasing, electron–electron scattering, electron-phonon coupling and lattice-heat dissipation, all of which take place at different timescales[14]. The observed instantaneous appearance within time resolution and decay of transient-absorption signal upon the excitation of ITO and ITO/SiO$_2$ are ascribed to electron dephasing and carrier scattering including electron–electron and electron-phonon scattering, respectively[14].

On the other hand, the ITO/SnO$_2$ and ITO/TiO$_2$ hetero-interfaces also showed the instantaneous appearance of signal within the time resolution, decaying dominantly with the time constant of 0.21 ± 0.1 ps and 0.14 ± 0.09 ps, respectively (Supplementary Tables 1 and 2). Subsequently, the signal becomes almost constant, which can be regarded as the component with lifetime much longer than time window of instrument (>3 ns). This instantaneous appearance within the time resolution and fast decay of signal is assignable to the above-mentioned LSPR-induced ultrafast events, and the long-lived component is assignable to FCA of SnO$_2$ and TiO$_2$, respectively[15–17]. To confirm FCA formation, we measured the IR spectra of the ITO/ SnO$_2$ interface at 10 ps after excitation by the 1700-nm laser

(Fig. 2b). The observed broad-absorption band from the near-IR to mid-IR region agrees well with the FCA of SnO$_2$[15]. This result strongly indicates that the hot electrons generated in ITO NCs were injected into the conduction band of SnO$_2$. As expected, no FCA was observed for SnO$_2$ without ITO NCs and ITO NCs alone, and an ITO/SiO$_2$ interface (Fig. 2a, Supplementary Fig. 7 and 8). For further confirmation of LSPR-induced carrier generation, we performed an excitation-spectrum measurement to probe the FCA. The excitation spectrum at 5000 nm and at 10 ps after the laser pulse reproduced the LSPR band of the ITO NCs, clearly proving LSPR-induced electron injection from ITO to SnO$_2$ (Fig. 2c). The possibility of non-linear optical phenomena caused by the enhanced electromagnetic field around the ITO NCs[18], was ruled out by the laser-power dependence of the FCA intensity at 5000 nm (see Supplementary Fig. 9). These facts clearly demonstrate that LSPR-induced electron injection proceeds from the ITO NCs to SnO$_2$.

Notably, as shown in Fig. 2c, the FCA was observed even under excitation of NIR-to-SWIR lasers (1000–2500 nm). The quantum yield (Φ) of LSPR-induced charge injection at an ITO/SnO$_2$ heterointerface was calculated by equation (1),

$$\Phi = \frac{n_e}{N_{photon}}, \qquad (1)$$

where $n_e$ (cm$^{-3}$) is the free-carrier number per unit volume injected into SnO$_2$ and $N_{photon}$ (cm$^{-3}$) is the absorbed-photon number of the pump light per unit volume in ITO/SnO$_2$. The electron-injection efficiency of the ITO/SnO$_2$ interface by excitation with a 1700-nm laser pulse was determined to be 33% from equation (1) (see Methods for details)[15]. Furthermore,

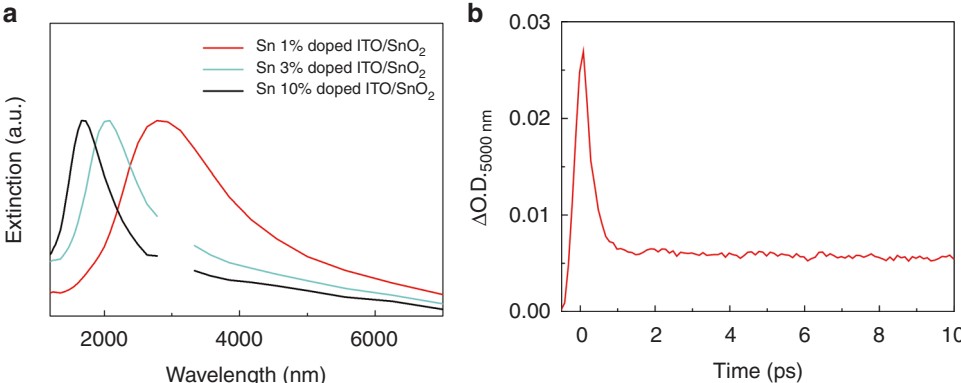

**Fig. 3** Extinction spectra and kinetic profiles. **a** Extinction spectra of heterointerfaces between ITO NCs with different Sn-doping ratios and $SnO_2$. The gaps at around 3000 nm of the 10%-Sn-doped ITO/$SnO_2$ or 3%-Sn-doped ITO/$SnO_2$ heterointerfaces mask noise from the instrument. **b** Kinetic profile of the 1%-Sn-doped ITO/$SnO_2$ after excitation with a 2500-nm laser (6 μJ/pulse) (FWHM of IRF in the system = 285 ± 40 fs)

we demonstrated that the responsive IR light of the ITO/$SnO_2$ system is extendable over 4000 nm. Since the peak position of LSPR of heavily-doped semiconductors is sensitive to the carrier density, the LSPR absorption of ITO can be easily tuned by changing the doping ratio of Sn[13]. As shown in Fig. 3, ITO NCs doped with 1% Sn, which exhibited LSPR absorption at around 2000–5000 nm, also showed LSPR-induced electron injection. This fact clearly indicates that IR light in the mid-wavelength infrared region (3000–8000 nm), which is the important region for the thermal sensors or free-space optical communications[19], can be applicable to the present system.

The FCA of ITO/$SnO_2$ monotonically decayed with a triple exponential function and the time constants are measured to be 2.0 ± 0.1, 33 ± 1 and 160 ± 1 μs (Fig. 2d and Supplementary Table 3). From the change of decay profiles depending on the size of $SnO_2$, we concluded that multiple decay profiles of FCA reflect the multiple decay channels of FCA, including the charge recombination and carrier trapping (see Supplementary Figs.11 and 13 for detail). The decay of FCA represents the lifetime of charge separation, which is an important parameter in determining the light-energy-conversion efficiency or quantum yield of optical detector. The long-lived charge separation is favourable for efficient energy conversion. For plasmonic carrier injection systems, fast charge recombination is a major obstacle facing efficient light-energy conversion[5,20]. The long-lived charge separation in our ITO/$SnO_2$ system, which reaches ~2–160 μs, indicates that combining plasmonic ITO with $SnO_2$ is a promising technique for IR-light sensors and energy-conversion systems. The photocurrent measurement of the ITO NCs/$SnO_2$/W photoelectrode further proves that IR-light-induced carrier injection from ITO to $SnO_2$ provides practicable electromotive force, even under irradiation with IR light (Fig. 4). Although the IPCE measurement succeeded up to 1600 nm due to the limitations of the instrument (Fig. 4b), the photocurrent was also successfully extracted through the external circuit, even under SWIR light (1615–2280 and 2093–2547 nm in Fig. 4c, d, respectively) from the Xe lamp.

## Discussion

It was revealed that the LSPR-induced electron injection from ITO to the $SnO_2$ or $TiO_2$ phases takes place, but the real mechanism should be identified because there are several possible mechanisms for the formation of FCA in metal oxides by the sensitisation of plasmonic NCs, i.e., plasmon-induced hot-electron injection[5], local-electromagnetic-field-induced in-situ electron and hole generation in a semiconductor[21] and plasmon-

induced resonant-energy transfer (PRET)[22]. PRET only occurs when the LSPR band of ITO NCs overlap with the absorption band of the semiconductor. However, the absorption of $SnO_2$ does not overlap with the LSPR of ITO. On the other hand, the enhanced local electromagnetic fields of plasmonic materials can contribute the generation of electron-hole pairs in the semiconductor, even if the interlayer effectively blocks the injection of hot electrons generated by LSPR excitation[21]. To clarify whether this mechanism is responsible for our system, we measured the TR-IR of ITO/oleylamine(OAm)/$SnO_2$, in which the ITO NCs are attached to the $SnO_2$ through the insulating OAm with a molecular length of ~2 nm. As the result, the FCA was not observed for the ITO/OAm/$SnO_2$, owing to the complete obstruction of electron injection from ITO NCs by insulating OAm layers (Supplementary Fig. 14). Consequently, we can conclude that LSPR-induced hot-electron injection is a key mechanism for the emergence of the FCA of $SnO_2$. Additionally, the appearance of FCA within a short timescale (~1 ps) strongly suggests that hot-electron injection takes place in the tunnelling process through the Schottky barrier at the ITO/$SnO_2$ heterointerfaces. Recently, the electron-injection mechanism from metal (Au or Al) to metal-oxide semiconductors upon LSPR excitation was investigated[23,24]. Both the LSPR-induced hot-electron injection and electron transfer via inter- or intraband transitions in metal were concluded to be responsible for electron injection. Note that the LSPR band of the ITO NCs was completely separated from the inter- of intraband-transition bands. This means that our ITO/$SnO_2$ exhibited efficiency of 33% by only selective-excitation of LSPR (without inter or intraband). This fact is favourable for the application of the present LSPR-induced carrier transfer system to the invisible optical devices. In contrast to ITO/$SnO_2$, the electron-injection efficiency to $TiO_2$ (anatase) was poor (0.11%).

The hot-electron-injection efficiency of ITO/$SnO_2$ is significantly high compared with that of ITO/$TiO_2$, implying that the Schottky barrier at the heterointerface is a criterion for inducing such injection. It was reported that the ITO has both semiconducting and metallic natures and forms the Schottky barrier at the ITO/$TiO_2$ interface[25]. The height and thickness of the Schottky barrier at the ITO/$TiO_2$ heterointerface should be larger than that at the ITO/$SnO_2$ heterointerface because the barrier is affected by the alignment of $E_F$ of materials forming these interfaces. Thus, in the case of the ITO/$TiO_2$ system, the hot electrons generated in the ITO are unfavourable for tunnelling through or overcoming the Schottky barrier at the heterointerface. This barrier is an obstacle to electron injection and can

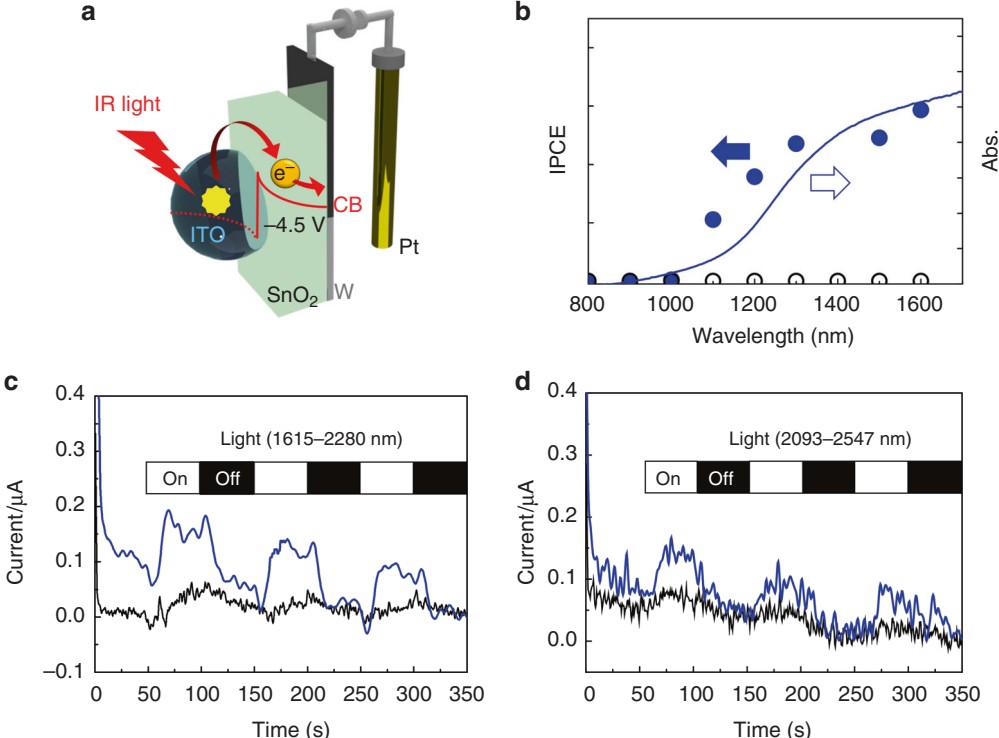

**Fig. 4** Illustration of IR-light-to-energy conversion and photoelectrochemical measurements. **a** The photoelectrochemical cell configuration and schematic illustration of IR-light-to-energy conversion using the plasmonic ITO/SnO$_2$ heterointerface (the reference electrode is omitted for clarity). **b** Action spectra of ITO-NC/SnO$_2$/W photoelectrodes. The filled and empty blue circles correspond to the ICPE of the ITO-NC/SnO$_2$/W electrode and the SnO$_2$/W electrode, respectively. The blue solid line is the diffuse-reflectance spectrum of the ITO-NC/SnO$_2$/W electrode. IPCE measurements under irradiation by light of wavelengths longer than 1600 nm was not carried out due to the limitations of our instrument. Experiments were performed in an acetonitrile solution of triethanolamine (10% v/v) containing 0.1-M tetrabutylammonium hexafluorophosphate. **c**, **d** Photoelectrochemical measurement of an ITO NCs/SnO$_2$/W electrode under SWIR irradiation. Short-circuit photocurrent and time (I-t) curves under irradiation by a chopped Xe lamp through a broadband pass filter (1615 nm–2280 nm, 104 mW cm$^{-2}$ or 2093 nm–2547 nm, 33 mW cm$^{-2}$) using a three-electrode setup (blue line). (Working electrode: ITO NCs/SnO$_2$/W electrode; Reference electrode: Ag$^+$/Ag; Counter electrode: Pt wire; Photoactive area: 2.5 × 1.5 cm$^2$; Solution: acetonitrile solution of triethanolamine (10% v/v) containing 0.1-M tetrabutylammonium hexafluorophosphate). The black line is an I-t curve obtained using the SnO$_2$/ W electrode as a working electrode

suppress charge recombination as a positive effect. Such recombination suppression is thought to contribute to the long-lived charge separation of the ITO/SnO$_2$ system. The present results provide us with fine tuning of the plasmonic material/semiconductor heterointerface to realise efficient charge separation, which to the best of our knowledge has not yet been established in plasmonic electron-injection systems.

In conclusion, we have demonstrated SWIR-LSPR-induced hot-electron injection from ITO NCs to SnO$_2$ or TiO$_2$ by means of the µs- and fs-TR-IR measurements. Considering the transparency of ITO and metal oxides, this is the first example of a clear and colourless system for IR-responsive carrier transfer, to the best of our knowledge. The selective-excitation of LSPR of ITO causes hot-electron injection with high efficiency (33%) and long-lived charge separation (~2–160 µs) thanks to fine control of the heterointerface. The reasonable electron injection efficiency and the much-longer-lived charge separation, compared with the typical Au/TiO$_2$ and other Au/semiconductor systems, guarantee the value of our system for optical devices using IR-light. Furthermore, we demonstrated that IR light longer than SWIR region can be applicable to the present system owing to the ease controllability of LSPR band of heavily doped semiconductor NCs. We believe that our experimental results provide an important step for the LSPR-induced carrier transfer and a useful strategy for use in invisible optical devices.

## Methods

**Transient-absorption measurements**. Microsecond (µs) time-resolved IR-absorption measurements were conducted using custom-built spectrometers, as described in our previous papers[26]. ITO/metal oxide samples were photoexcited by using a 1400 nm laser pulses (energy: 2.7 mJ pulse$^{-1}$, duration: 6 ns, repetition rate: 1 Hz) originating from a Nd: YAG laser (Continuum Surelite II) equipped with an optical parametric oscillator (OPO) system to generate the desired pump wavelength. The IR light emitted from the MoSi$_2$ coil was used as the probe light in the mid-IR region (7000–1000 cm$^{-1}$). The transmitted IR light from the ITO/metal oxide samples fixed on the CaF$_2$ plate was then introduced into the grating spectrometer and the monochromated light from the spectrometer was detected by an MCT detector (Kolmar), and then the output electric signal was amplified using an AC-coupled amplifier (Standford Research System SR560, 1 MHz). The time resolution of the spectrometers was limited to ~1 µs by the bandwidth of the amplifier. The instrument response function (IRF) was evaluated by measuring the scattered laser pulses detected by the MCT. The FWHM value of IRF was estimated to be 0.485 µs as indicated in Supplementary Figure 12.

In the femtosecond-to-picosecond region, the ultrafast kinetic measurements were performed using on Ti:sapphire laser system (Spectra Physics, Solstice and TOPAS Prime, duration: 90 fs, repetition rate: 1 kHz) to generate the pump and probe wavelengths[27]. The ITO/metal oxide samples were photoexcited using 1700 nm (energy: 6 µJ pulse$^{-1}$). The probe light was focused on the sample and the transmitted IR light during irradiation condition entered the spectrometer equipped with gratings. The monochromated light was then detected by MCT detector. The FWHM of IRF value was estimated to be 285 ± 40 fs (refer to Figure S6b). For the measurement of the kinetic profile shown in Supplementary Fig. 14, a femtosecond Ti:sapphire laser system (Spectra Physics; Hurricane and TOPAS; wavelength: 800 nm; pulse duration: 150 fs; repetition rate: 1 kHz) was used. The FWHM value of IRF of the system is 210 fs. The 1700-nm pulse from one OPA was used as a pump light. For the probe light, a 3440-nm pulse generated from the other OPA with a difference-frequency-generation crystal was used. The

intensity of the probe light transmitted from the sample was detected using an MCT photodetector (KMPV11-1-J1, Kolmar technology).

**Transmission electron microscopy**. TEM observations (JEM-1011, JEOL) were carried out at an accelerating voltage of 100 kV. TEM samples were prepared by placing a drop of cluster solution onto a carbon-coated copper grid. HAADF-STEM and EDS elemental mapping were performed on Titan[3] (FEI) (convergence semiangle: 17.9 mrad, a high-brightness Schottky emission gun (X-FEG) and double spherical aberration correctors, Oxford Instruments X-Max$^N$ 100TLE EDS detector) electron microscopes with an operating voltage of 300 kV.

**Spectrometry**. Steady-state UV–vis–NIR-absorption spectroscopy was conducted using a U-4100 spectrophotometer (HITACHI).

**X-ray diffraction analysis**. XRD patterns were taken by X'pert Pro MPD (PANalytical) with CuKα radiation (λ = 1.542 Å) at 45 kV and 40 mA.

**Synthesis of ITO nanocrystals**. ITO NCs doped with different ratio of Sn was synthesised as follows[13]. An $n$-octylether (10-mL) suspension of indium(III) acetate (1.2–$x$ mmol), tin(II) 2-ethylhexanoate ($x$ mmol), 2-ethylhexanoic acid (3.6 mmol) and oleylamine (10 mmol) was stirred at 80 °C under vacuum for 30 min. The solution was heated at 150 °C for 1 h under a $N_2$ atmosphere and stirred for a further 2 h at 280 °C to afford the formation of ITO NCs. After cooling to room temperature, oleic acid (10.8 mmol) was injected into the solution and then stirred for 30 min under a $N_2$ atmosphere. Repeated centrifugal purification by ethanol yielded pure ITO NCs protected by oleic acid. Finally, ITO NCs were re-dispersed in chloroform.

**Loading of ITO nanocrystals onto metal oxides**. ITO NCs were adsorbed onto nano-sized oxides through the dipole-induced dipole or dipole−charge interactions between the hydrophobic ITO NCs and hydrophilic oxides[28]. We used $SiO_2$ (G-10, Fuji Sylisia), $TiO_2$ (anatase or P-25, Aldrich) and $SnO_2$ (Wako, 22–43 nm) as oxide supports. Oxide supports (100 mg) were added into the chloroform solution (50 mL) containing the desired quantity of ITO NCs (10-wt%-Sn doping vs oxides, wt of oxides were calculated as the sum of In and Sn), followed by stirring for 24 h. The solution was filtered and washed with hexane and chloroform. Based on the absorbance change of the filtrates before and after stirring with oxides for 24 h, we concluded that nearly all the ITO NCs were adsorbed onto oxide supports. After the adsorption, the sample was putted on the glass plate and calcined in air at 600 °C for 30 min to remove organic compounds, while the intensity of the LSPR peak was significantly reduced in all ITO/metal oxides upon the calcination due to the decrease in the free-electron density. This calcination removed the surface ligands of ITO NCs, thereby creating a heterointerface between ITO NCs and oxide supports. After calcination, the ITO/metal oxides are annealed under reductive atmosphere (4% $H_2$/Ar) at 280 °C for 5 h to regenerate the LSPR. TEM and XRD measurements revealed no changes in size and crystalline structure of the ITO/oxide heterointerfaces upon heat treatments (Supplementary Fig. 1 and 2). For the measurement of absorption spectrum in IR region, we used $CaF_2$ as substrate. All ITO/oxide show an LSPR peak in the NIR region, even after annealing under reductive atmosphere.

**Preparation of the ITO-NC/SnO$_2$/W photoelectrode**. A dense $SnO_2$ layer was prepared on tungsten substrates (Nilaco, 1.5 cm × 3 cm) by spin-coating 1-butanol containing 0.1-M $SnCl_2 \cdot 2H_2O$ (Aldrich) and annealing at 450 °C for 30 min. A mesoporous $SnO_2$ layer was formed on the dense $SnO_2$ layer using the $SnO_2$-Sol squeeze method (obtained by mixing 10 mL of 1-butanol with 1.0 g of $SnO_2$ NPs (diameter = 22–43 nm, Wako), 132.4 mg of ethylcellulose (90–110 mPa·s, TCI) and 1 mL of acetylacetone (Wako)), followed by annealing at 450 °C for 30 min. The thickness of the $SnO_2$ film, which was measured by scanning electron microscopy (S-4800, HITACHI), was 5 μm. ITO NCs were deposited onto the surfaces of the $SnO_2$ films by spin-coating an octane solution containing ITO NCs (50 mg mL$^{-1}$). The resulting electrode was calcined in air at 600 °C for 30 min, followed by anneal under 4% $H_2$/Ar at 280 °C for 5 h.

**Estimation of the quantum yield of electron injection at an ITO/SnO$_2$ heterointerface**. The quantum yield ($\Phi$) of LSPR-induced charge injection at an ITO/$SnO_2$ heterointerface was calculated by equation (1),

$$\Phi = \frac{n_e}{N_{photon}}, \quad (1)$$

where $n_e$ (cm$^{-3}$) is the free-carrier number per unit volume injected into $SnO_2$ and $N_{photon}$ (cm$^{-3}$) is the absorbed-photon number of the pump light per unit volume in ITO/$SnO_2$.

$N_{photon}$ was calculated by equation (2),

$$N_{photon} = \frac{A \times \text{Total energy of one pulse}}{\text{Energy of single photon}} \times (1 - 10^{-\text{O.D.@1700}})/V, \quad (2)$$

where $A$ is the ratio of the beam intensity within the FWHM of a laser pulse (0.5)

to the full intensity assuming that the laser pulse has a Gaussian intensity profile. The total energy of one pulse is 1 μJ and the single-photon energy was estimated as

$$\text{Energy of single photon} = hc/\lambda, \quad (3)$$

where $h$, $c$ and $\lambda$ are the Planck constant (6.626 × 10$^{-34}$ J·s), the speed of light (2.998 × 10$^8$ m/s) and the wavelength of the pump laser (1700 nm), respectively. From equation (3), the single-photon energy was calculated as 1.17 × 10$^{-19}$ J. $V$ is the volume of the pump-laser path in the sample pellet and is calculated to be 3.75 × 10$^{-6}$ cm$^3$ from the FWHM of the pump laser (349 μm) and the optical-path length ($l$ = 39.2 μm). The OD of the sample is 1.35 at the excitation wavelength (1700 nm). Thus, $N_{photon}$ was calculated as 0.109 × 10$^{19}$ cm$^{-3}$. The absorption coefficient, $\alpha$ (cm$^{-1}$), due to free-carrier injection into the $SnO_2$ was calculated according to equation (4)

$$10^{-\Delta\text{O.D.@5000 nm}} = e^{-\alpha \cdot l}, \quad (4)$$

where $\Delta$OD at 5000 nm is 0.00160. From equation (4), $\alpha$ was calculated to be 0.939 cm$^{-1}$. Since the absorption cross section of the free carrier is expressed as $\sigma = \alpha/n_e$ (cm$^2$), $n_e$ is expressed by the following equation,

$$n_e = \frac{\alpha}{\sigma'} \quad (5)$$

where $\sigma'$ is the apparent absorption cross section of $SnO_2$ powder (2.59 × 10$^{-18}$ cm$^2$), which is calculated in the next section. From equation (5), $n_e$ was calculated to be 0.0363 × 10$^{19}$ cm$^{-3}$. Finally, the $\Phi$ value of electron injection was determined to be 33% from equation (1).

**Estimation of the apparent absorption cross section of SnO$_2$ powder**. According to ref. [15], the absorption cross section ($\sigma$) of $SnO_2$ is expressed as

$$\log(\sigma/10^{-18}[\text{cm}^2]) = a + b \cdot \log_{10}(\lambda[\text{nm}]), \quad (6)$$

where $a_\perp = -8.888$, $b_\perp = 2.894$, $a_{//} = -8.705$ and $b_{//} = 2.863$. When $\lambda$ is 5000 nm, $\sigma_\perp$ and $\sigma_{//}$ are estimated as 65.59 × 10$^{-18}$ cm$^2$ and 76.763 × 10$^{-18}$ cm$^2$, respectively. As the direction of a-axis and b-axis is equal, the averaged $\sigma_{ave}$ value was calculated to be 69.312 × 10$^{-18}$ cm$^2$ using the equation $\sigma_{ave} = (\sigma_{//} + 2\sigma_\perp)/3$. Since the $\sigma_{ave}$ value is the value of a single $SnO_2$ crystal, a correction that takes the number density of $SnO_2$ powder into account is necessary to evaluate the apparent absorption cross section ($\sigma'$). As the density of single-crystal $SnO_2$ and the bulk density of the sample powder are 6.95 g/cm$^3$ and 0.26 g/cm$^3$, respectively, $\sigma'$ was calculated by using the following equation:

$$\sigma' = (0.26/6.95) \cdot \sigma_{ave.} \quad (7)$$

From equation 6, $\sigma'$ was calculated as 2.593 × 10$^{-18}$ cm$^2$.

**Estimation of the quantum yield of electron injection at an ITO/TiO$_2$ (anatase) heterointerface**. The quantum yield ($\Phi$) of LSPR-induced charge injection at an ITO/$TiO_2$ (anatase) heterointerface was estimated using the same procedure as in the case of the ITO/$SnO_2$ interface discussed above.

In the present case, the total energy of one pulse is 6 μJ and the energy of a single photon was 1.17 × 10$^{-19}$. The volume of the pump-laser path in the sample pellet ($V$) was calculated to be 5.55 × 10$^{-6}$ cm$^3$ based on the FWHM of the pump laser (349 μm) and the optical-path length ($l$ = 58 μm). The OD of the sample is 1.49 at the excitation wavelength (1700 nm). Thus, $N_{photon}$ is calculated to be 4.48 × 10$^{18}$ cm$^{-3}$. The absorption coefficient, $\alpha$ (cm$^{-1}$), due to the injection of free carriers into $TiO_2$ (anatase) was estimated to be 0.635 cm$^{-1}$ from the $\Delta$OD at 5000 nm (=0.00160) and the optical-path length $l$ (=58 μm). The apparent absorption cross section ($\sigma'$) of $TiO_2$ (anatase) powder is 1.303 × 10$^{-16}$ cm$^2$, which is determined in next section. From the values of $\alpha$ and $\sigma'$, $n_e$ was calculated to be 4.87 × 10$^{15}$ cm$^{-3}$. Finally, the $\Phi$ of electron injection was estimated to be 0.11% at the ITO/$TiO_2$ (anatase) heterointerface.

**Estimation of the apparent absorption cross section of TiO$_2$ (anatase) powder**. According to refs. [16] and [29], the average $\sigma_{ave}$ value was calculated to be 6.549 × 10$^{-16}$ cm$^2$. As the $\sigma_{ave}$ value is that of a single crystal of $TiO_2$ (anatase), correction taking the number density of $TiO_2$ powder into account is necessary to evaluate the apparent absorption cross section ($\sigma'$). Since the density of single-crystal $TiO_2$ (anatase) and the bulk density of the sample powder ranges from 3.90 g/cm$^3$ to 0.776 g/cm$^3$, respectively, $\sigma'$ was calculated as

$$\sigma' = (0.776/3.90) \cdot \sigma_{ave.} \quad (8)$$

From equation 8, the $\sigma'$ value of $TiO_2$ (anatase) was calculated to be 1.303 × 10$^{-16}$ cm$^2$.

## Data availability

The data that support the findings of this study are available from the corresponding author upon reasonable request.

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

## Acknowledgements

We thank T. Nagai, and K. Kimoto for the observation of HAADF-STEM, HRTEM image and EDS mapping. This work was supported by JSPS KAKENHI grant number JP16H06520 (Coordination Asymmetry) (T.T.), JP17H05257 (Photosynergetics) (M.S.), JP17H05491 (Mixed Anion) (A.Y.), JP17K19031 (Grant-in-Aid for Challenging Exploratory Research) (M.S.), the Collaborative Research Project of the Institute of Chemical Research, Kyoto University and AIST Nano-characterisation Facility platform as a programme of "Nanotechnology Platform" of the Ministry of Education, Culture, Sports, Science and Technology (MEXT), Japan. We thank T. Yoshinaga for the support of synthesising samples and experiments. This work was supported by NIMS microstructural characterization platform as a program of "Nanotechnology Platform" of MEXT, Japan.

## Author contributions

M.S. conceived and designed the experiment. T.T. supervised the study and provided intellectual and technical guidance. M.S., M.K. and T.Y. synthesised the ITO NCs and ITO/semiconductor oxides. T.K. and M.K. fabricated the photoelectrodes and performed the photoelectrochemical experiments. J.J.M.V., H. Matsunaga., C.S.K.R. and A.Y. carried out the fs- and μs- TR-IR experiments. A.F. and H. Matsuzaki carried out the fs-TR-IR experiments shown in Supplementary Fig. 14. M.S. wrote the manuscript. All authors participated in discussion of the research.

## Additional information

**Competing interests:** The authors declare no competing interests.

