## [Peer Review File · Nature Communications]

Reviewers' comments:

Reviewer #1 (Remarks to the Author):

In this paper, the authors developed novel technics to greatly extend the charge carrier dynamics characterization in invisible region using complementary set of IR pump probe methods. I believed it can definitely provide large contribution to the community, specially to the extend the methodology of the photophysical studies for invisible samples. However, I still have some major concerns in terms of the data interpretation and conclusion. They should be solved before it can be considered publishing.

1- For the ITO NCs on different MO_x, the author mentioned glass as substrate which will not fit with the IR measurements. Regarding the film fabrication, it is not clear for me what do the author mean by "weak interactions between the hydrophobic ITO NCs and hydrophilic oxides" in the methods part.

2- In figure 1C, the LSPR is broader when the ITO is attached to SnO₂, why? Is ITO has different size on SnO₂ compare to the others MO_x? Another required information is needed about the ITO size and size distribution?

3- in figure S2, the author said that there is "No coalescence, aggregation, or change in size of the ITO NCs were observed before and after thermal and reductive annealing" which for the reader eyes looks very difficult. Can the author add more analysis to the TEM images at least to be comparable to the ITO.

4- I think it will be more easy to follow the kinetic traces if the time axis in Log scale.

5- At line 119, the figure needs to change from 2d to 2c.

6- In figure 2b, if the author collect the spectra from different traces they need to make it with small intervals. I do not know the point behind adding this figure where there is no features to show. Please remove it or clearly the point behind it.

7- "PRET needs to overlap with the LSPR of ITO and the absorption of SnO₂, although the absorption of SnO₂ does not overlap with the LSPR of ITO", this needs to be more clear.

Reviewer #2 (Remarks to the Author):

The authors report on the optical properties of ITO/MeO₂, where Me is Ti, Sn and Si and which are transparent in the visible spectral range. The focus of the study is made on the evaluation of transfer of the hot electron induced by plasmon to the e.g. SnO₂. The efficiency of the electron transfer is estimated to be around 33%, that is rather good. Overall it is a good study. However, it contains a number of confusing statements. First, the authors stress the applicability of their systems for telecommunication. It is not obvious (to me) why telecommunication needs transparency in the visible spectral range (what is wrong with standard diodes, or PbSe). The transparency can be important though for photovoltaic application (e.g. photochromic windows, etc) Second, the best performing system is ITO/SnO₂. I would like to recommend removal to SI all information related to TiO₂ and SiO₂.

Third, the figure captions are very long and hard to follow as well as introduction. Please, remove referral too natural photosynthetic systems, that are not relevant here.

Importantly, please, define the focus of your study and edit your manuscript accordingly.

Reviewer #3 (Remarks to the Author):

The authors demonstrate electron transfer into transparent metal oxides (TiO₂, SnO₂) from irradiation of ITO nanocrystals in the NIR. They find a high efficiency for the hot-electron injection, 33%, into SnO₂, with lower efficiency into TiO₂ and report very long-lived charge separation states for the system. This is a useful system with potential applications in the NIR region of the spectrum and the results will be of interest to others in the community and the wider field. However, some areas of the manuscript require major revision prior to publication of the work.

The sections relating to the time-resolved measurements needs significant revision. The experiments were carried out on instruments with fs pulses, however no IRF (instrument response function) is included in the data or experimental section for these experiments. In addition, it is noted that the decay of the plasmon is within 1 – 2 ps. For slower, ns flash photolysis, the time constants are “estimated” as 1.8, 28 and 120 ns, however again no IRF is shown. All decay constants should be accurately reported along with their errors, with fitting with convolution to account for the IRF. Instrument IRFs should be both mentioned in the experimental section and shown on the kinetic traces.

For faster timescales (fs/ps), a discussion of the accurately measured plasmon decay should be included and compared to the timescales of the different hot-electron decay processes (occurring in ITO alone). The differences in lifetime of the decay of the ITO plasmon resonance both in the presence and absence of the SnO₂/TiO₂ should be noted and fully included in the analysis/discussion.

For slower timescales, along with accurately reporting the decay constants, further discussion on the nature of the different observed lifetimes is highly recommended. Currently, these are attributed to “the nonuniformity of SnO₂ powder”. However, there is no quantification of the particle size distribution in the SnO₂ powder either before or after this statement and so the evidence behind this statement is unclear. Presumably also, changing the particle size distribution of the SnO₂ powder would change the lifetime of the charge-separated states and this is an experiment that could be carried out as proof of the statement. As the long-lived charge-separated state is one of the key results of the manuscript, it is important these issues are clarified.

I would also request more electron microscopy images of the heterointerfaces formed and investigated, both to allow consistency between areas of the structures to be established, as well as some at higher magnification showing the interfacial area.

June 6th, 2018

A point-by-point response to the reviewers' comments for NCOMMS-18-04874-A

RE: Nature Communications Manuscript Revision Request

Manuscript ID: NCOMMS-18-04874-A

Title: " Clear and Transparent Nanocrystals for IR-Responsive Carrier Transfer "

Author(s): Masanori Sakamoto,* Tokuhisa Kawawaki, Masato Kimura, Junie Jhon M. Vequizo, Hironori Matsunaga, Chandana Sampath Kumara Ranasinghe, Akira Yamakata, Hiroyuki Matsuzaki, Akihiro Furube, Toshiharu Teranishi*

Corresponding author's mailing address: Institute for Chemical Research, Kyoto University, Uji, Kyoto 611-0011, Japan

Corresponding authors' e-mail address: sakamoto@scl.kyoto-u.ac.jp; teranisi@scl.kyoto-u.ac.jp

Reviewers' comments:

Reviewer #1 (Remarks to the Author):

In this paper, the authors developed novel technics to greatly extend the charge carrier dynamics characterization in invisible region using complementary set of IR pump probe methods. I believed it can definitely provide large contribution to the community, specially to the extend the methodology of the photophysical studies for invisible samples. However, I still have some major concerns in terms of the data interpretation and conclusion. They should be solved before it can be considered publishing.

Response: We appreciate your comments and positive feedback. We have addressed all the comments. We hope that the explanation and revision of our work are clear enough.

1- For the ITO NCs on different MOx, the author mentioned glass as substrate which will not fit with the IR measurements.

Response: Thank you for your indication. We used the glass substrate for the uniform calcination of ITO/SnO₂. For the IR measurement, we used CaF₂ substrate.

Action: We revised the section of “Loading of ITO NCs onto metal oxides” and added sentence “For the measurement of absorption spectrum in IR region, we used CaF₂ as substrate.” in METHOD to avoid the misleading.

Regarding the film fabrication, it is not clear for me what do the author mean by "weak interactions between the hydrophobic ITO NCs and hydrophilic oxides" in the methods part.

Response: We used the method reported in *J. Am. Chem. Soc.* 128, 14278-14280 (2014) by Zheng et al. They did not identify the weak interaction, while they discussed that the weak interaction between oxide particles and hydrophobic metal nanoparticles is most likely due to dipole–induced dipole or dipole–charge interactions.

Action: We revised the sentence "weak interactions between the hydrophobic ITO NCs and hydrophilic oxides" to “dipole–induced dipole or dipole–charge interactions between the hydrophobic ITO NCs and hydrophilic oxides” the section of “Loading of ITO NCs onto metal oxides” in METHOD of revised version of manuscript.

2- In figure 1C, the LSPR is boarder when the ITO is attached to SnO₂, why?

Response: We consider that the plasmonic coupling between ITO NCs is responsible for the broadened LSPR band. According to the HRTEM image in Figure R1, some ITO NCs are closely immobilized on the surface of SnO₂.

Action: We added the sentence “The broadening of LSPR of ITO NCs on SnO₂ results from the plasmonic coupling of ITO NCs (see supplementary Fig. 5)” in the page 3, last sentence of second paragraph in the revised version of manuscript. We added the HRTEM, HAADF-STEM and EDS elemental mapping images in supplementary Fig. 5 in the revised version of manuscript.

Figure R1. **a**, HAADF-STEM image of ITO/SnO₂ hetero interface. **b**, HAADF-STEM-EDS elemental mapping images of ITO/SnO₂ hetero interface. **c**, HRTEM image of ITO/SnO₂ hetero interface.

Is ITO has different size on SnO₂ compare to the others MOx? Another required information is needed about the ITO size and size distribution?

Response: The as-synthesized ITO NCs are uniform in size, as shown in Fig. 1b inset. In addition, we also added the size distribution of ITO NCs on SnO₂ (Figure R2) according to your indication.

Action: We added the size distribution of ITO NCs and SnO₂ NCs in supplementary Fig. 3 in the revised version of manuscript.

Figure R2. Size distribution of **a** ITO NCs and **b** SnO₂.

3- in figure S2, the author said that there is "No coalescence, aggregation, or change in size of the ITO NCs were observed before and after thermal and reductive annealing" which for the reader eyes looks very difficult. Can the author add more analysis to the TEM images at least to be comparable to the ITO.

Response: According to your indication, we added the distribution of the size of ITO nanoparticle before and after deposition on the metal oxide (Figure R3). No size distribution change was observed.

Action: We added the size distribution of ITO NCs on SnO₂ before and after calcination in supplementary Figure 4 in the revised version of manuscript.

Figure R3. Size distributions of ITO NCs before and after calcination.

4- I think it will be more easy to follow the kinetic traces if the time axis in Log scale.

Response: According to your suggestion, we changed the kinetic traces in Log scale. We only showed the time axis in kinetic traces in ps region (Figure 2a, 3b, supplementary Figs. 6a, 8, and 12) as liner scale, because the liner axis seems to express the difference of intensity of FCA more clearly in these Figures.

Action: We change the kinetic traces in Fig. 2d, and supplementary Figs. 11S in log scale in the revised version of manuscript.

5- At line 119, the figure needs to change from 2d to 2c.

Response: Thank you very much for your indication.

Action: We corrected the Fig. 2d to 2c at page 4, line 14 of third paragraph in the revised version of manuscript.

6- In figure 2b, if the author collect the spectra from different traces they need to make it with small intervals. I do not know the point behind adding this figure where there is no features to show. Please remove it or clearly the point behind it.

Response: Thank you for indication. Due to the limitation of our instrument, it is extremely difficult to take the data with small intervals. Therefore, we added the theoretically calculated curve (red line) in Figure 2b to reinforce our claim.

Figure 2b is important to confirm that the observed transient species is a free carrier in SnO₂ (i.e. FCA). The FCA of semiconductor shows such structureless feature rising to the longer wavelength (for example, *Appl. Phys. Lett.* **100**, 011914 (2012)). Since the observed broad-absorption band from near-IR to mid-IR region agreed well with the theoretically fabricated FCA shape of SnO₂, we

concluded that the hot electrons generated in ITO NCs were injected into the conduction band of SnO₂.

Action: We added the following sentence to clarify the meaning of Fig. 2b. The observed broad-absorption band from near-IR to mid-IR region agreed well with the FCA of SnO₂.¹³ "This result strongly indicates that the hot electrons generated in ITO NPs injected into the conduction band of SnO₂." in the page 4, line 8 of third paragraph in the revised version of manuscript.

7- "PRET needs to overlap with the LSPR of ITO and the absorption of SnO₂, although the absorption of SnO₂ does not overlap with the LSPR of ITO", this needs to be more clear.

Response: We changed the unclear description of PRET in the revised version of manuscript.

Action: We revised the sentence as follows: "PRET only occurs when the LSPR band of ITO NCs overlaps with the absorption band of semiconductor. However, there is no spectral overlap between LSPR and SnO₂." in the page 6, line 6 of first paragraph in the revised version of manuscript.

Reviewer #2 (Remarks to the Author):

The authors report on the optical properties of ITO/MeO₂, where Me is Ti, Sn and Si and which are transparent in the visible spectral range. The focus of the study is made on the evaluation of transfer of the hot electron induced by plasmon to the e.g. SnO₂. The efficiency of the electron transfer is estimated to be around 33%, that is rather good. Overall it is a good study. However, it contains a number of confusing statements.

Response: We appreciate your comments and positive feedback. We have addressed all the comments. We hope that the explanation and revision of our work are clear enough.

First, the authors stress the applicability of their systems for telecommunication. It is not obvious (to me) why telecommunication needs transparency in the visible spectral range (what is wrong with standard diodes, or PbSe). The transparency can be important though for photovoltaic application (e.g. photochromic windows, etc)

Response: We agree with your suggestion.

Action: We deleted the sentence about telecommunication from the introduction of revised version of manuscript.

Second, the best performing system is ITO/SnO₂. I would like to recommend removal to SI all information related to TiO₂ and SiO₂.

Response: Thank you for your indication

Action: We moved the information related to TiO₂ and SiO₂ to supplementary Fig. 6 in the revised version of manuscript.

Third, the figure captions are very long and hard to follow as well as introduction. Please, remove referral too natural photosynthetic systems, that are not relevant here.

Response: We revised the introduction part and figure captions.

Action: We deleted following sentence “Various colourful pigments have evolved in natural photosystems for harvesting light” from the introduction of the revised version of manuscript. In addition, we made the figure captions simpler in the revised version of manuscript.

Importantly, please, define the focus of your study and edit your manuscript accordingly.

Response: Thank you for your suggestion.

Action: We defined the focus of our study and edit our manuscript.

Reviewer #3 (Remarks to the Author):

The authors demonstrate electron transfer into transparent metal oxides (TiO₂, SnO₂) from irradiation of ITO nanocrystals in the NIR. They find a high efficiency for the hot-electron injection, 33%, into SnO₂, with lower efficiency into TiO₂ and report very long-lived charge separation states for the system. This is a useful system with potential applications in the NIR region of the spectrum and the results will be of interest to others in the community and the wider field. However, some areas of the manuscript require major revision prior to publication of the work.

Response: We appreciate your comments and positive feedback. We have conducted additional experiments and to address all the comments. We hope that the explanation and revision of our work are clear enough.

The sections relating to the time-resolved measurements needs significant revision. The experiments were carried out on instruments with fs pulses, however no IRF (instrument response function) is included in the data or experimental section for these experiments. In addition, it is noted that the decay of the plasmon is within 1 - 2 ps. For slower, us flash photolysis, the time constants are "estimated" as 1.8, 28 and 120 us, however again no IRF is shown. All decay constants should be accurately reported along with their errors, with fitting with convolution to account for the IRF. Instrument IRFs should be both mentioned in the experimental section and shown on the kinetic traces.

Response: The IRFs of fs-LFP system and μ s-LFP are c.a. 100 fs and ca. 2 μ s, respectively. The time constant of c.a. 2 μ s estimated by the curve fitting of kinetic traces in μ s region corresponds to the FCA affected by IRF of μ s LFP. In addition, we re-analyzed the kinetic profiles taking the IRF into accounts.

Action: We added IRFs to the caption of Fig 2 and 3 and the section of "Transient-absorption measurement" in METHOD in the revised version of manuscript. In addition, we re-analyzed kinetic profiles taking the IRF into the consideration.

For faster timescales (fs/ps), a discussion of the accurately measured plasmon decay should be included and compared to the timescales of the different hot-electron decay processes (occurring in ITO alone). The differences in lifetime of the decay of the ITO plasmon resonance both in the presence and absence of the SnO₂/TiO₂ should be noted and fully included in the analysis/discussion.

Response: Thank you for your important indication. We estimated the decay rates of LSPR in ITO/metal oxides in ps region. The lifetime of hot electrons in ITO/SiO₂ is longer than those in ITO/other metal oxides. This difference corresponds the low thermal conductivity of SiO₂

in comparison with the other dielectric mediums, because the thermal conductivity of SiO₂ is quite small in comparison with other metal oxide (Table R1). Hamanaka, Y. *et al.*, reported that the thermal conductivity is one of the important parameter governing the relaxation dynamics of Au NCs. According to their research, the matrix of low thermal conductivity, such as SiO₂, significantly suppresses the relaxation and extend the lifetime of excited Au NCs (*Phys. Rev. B*, **63**, 104302, (2001).). We consider that the similar phenomenon was observed in the present system. We added this discussion in the supplementary information.

Action: We added the estimation of the decay rates of ITO plasmon with different substrates in the supplementary Fig. 5 and Table 1 of supporting information. In addition, we gave the discussion in the section of “Investigation on the decay profiles of ITO/metal oxide upon the excitation of LSPR in ps region” in the supplementary information in the revised manuscript.

Figure R4. Kinetic profiles of transient absorption of ITO and ITO/metal oxides at the ps scale at 5,000 nm upon the excitation by a 1,700-nm laser (instrument response function (IRF): c.a. 100 fs). The kinetic profiles were analysed using a convolution method using IRF as a gaussian curve. The red lines are best fit.

Table R1. Decay constants of LSPR in the ps region.

	Thermal conductivity of metal oxides (W/mK)	τ (ps)
ITO/SiO ₂	1.3 ¹	0.43±0.02
ITO/SnO ₂	10 ²	0.28±0.04
ITO/TiO ₂	11 ³	0.21±0.01

¹ Andersson, S. & Dzhavadov, L. Thermal conductivity and heat capacity of amorphous SiO₂: pressure and volume dependence, *Journal of Physics: Condensed Matter*, **29**, 6209-6216 (1992). ² Li, S. *et al.*, Thermal conductivity of individual tin dioxide nanobelt, *Appl. Phys. Lett.* **84**, 2638, (2004). ³ Touloukian, Y. S. Powell, R. W. Ho, C. Y. & Klemens, P. G. *Thermophysical Properties of Matter - The TPRC Data Series. Volume 2. Thermal Conductivity - Nonmetallic Solids*, 1971.

For slower timescales, along with accurately reporting the decay constants, further discussion on the nature of the different observed lifetimes is highly recommended. Currently, these are attributed to "the nonuniformity of SnO₂ powder". However, there is no quantification of the particle size distribution in the SnO₂ powder either before or after this statement and so the evidence behind this statement is unclear. Presumably also, changing the particle size distribution of the SnO₂ powder would change the lifetime of the charge-separated states and this is an experiment that could be carried out as proof of the statement. As the long-lived charge-separated state is one of the key results of the manuscript, it is important these issues are clarified.

Response: Thank you for indication. According to your advice, we challenged to investigate the charge separation process between ITO and different-sized SnO₂ particles. Since we can obtain two types of commercial SnO₂ NCs (SnO₂(S) (diameter:27±13 nm) and SnO₂(L)(diameter:276±155 nm) (Fig. R5), we tried to investigate the carrier dynamics of ITO/SnO₂(S) and ITO/SnO₂(L) (Fig. R6). The time-resolved IR spectrum of ITO/SnO₂(S) and ITO/SnO₂(L) after the excitation LSPR band of ITO are shown in Fig. R7. At 5 μs after excitation, the spectrum of overlapping of FCA and trapped carrier peaks at around 3800 nm was observed. We observed the clear FCA spectrum in ps region (Figure 2b in the main text to the shape of FCA of SnO₂). Therefore, this result indicated that the FCA in SnO₂ generated by the electron injection from ITO NCs decay via charge recombination and trapping. Fig. R8 shows the decay profile of transient absorption at 5000 nm. The decay profiles of ITO/SnO₂(S) and ITO/SnO₂(L) were fitted well with the tri-exponential decay function (Table R2). Since the fast component is similar to the IRF of μs-transient absorption measurement system, it can be assigned to the FCA affected by IRF. The middle and the slowest components increased with increasing size of SnO₂. These results indicate that the multiple decay kinetics depending on the size of SnO₂ could correspond to the different diffusion process of electron concerning (i) the different sizes of SnO₂ and (ii) different defects trapping and de-trapping processes on SnO₂ (*J. Photochem. Photobiol. C* **15**, 21-30, (2013). and *J. Phys. Chem. C* 2015, 119, 24538.). Although it is very difficult to estimate the strict lifetime of charge separation from the result, the observed shape of transient spectrum indicates that the charge recombination of FCA to ITO concerns the decay process in the μs region. In addition, since the trapping state in the ITO/SnO₂ system estimated by the absorption edge is 0.25 eV from conduction bands minimum, the electron in the shallow trapped state could be retrieved as external energy (For example, *J. Phys. Chem. C* 2015, 119, 24538.).

Action: We added the experiment of size dependent decay profiles of ITO/SnO₂ in the supplementary information in the revised version of manuscript. In addition, we changed the sentence “The multiple decay profiles of FCA may reflect the nonuniformity of SnO₂ powder.” to “From the shift of decay profiles depending on the size of SnO₂, we concluded that multiple decay profiles of FCA reflect the multiple decay channels of FCA, including the charge recombination and carrier trapping.” in the page 5, line 4 of second paragraph in the revised version of manuscript. In addition, we added the discussion about the size dependent decay profiles of ITO/SnO₂ in the section of “Investigation on the decay profiles of ITO/SnO₂ with different sizes upon the excitation of LSPR in μ s region” in the supplementary of the revised version of manuscript.

Figure R5. a. TEM images of different sized SnO₂ NCs (SnO₂(S) and SnO₂(L)). **b.** Size distributions of SnO₂(S) and SnO₂(L).

Figure R6. TEM images of ITO NCs on SnO₂(L) during the immobilization process.

Figure R7. Time-resolved IR spectra of **a** ITO/SnO₂(S) and **b** ITO/SnO₂(L). The dipping feature from 2600 to 3600 nm indicated by black arrows is attributed to water (For example, *Applied Optics*, 12, 555-563 (1973)).

Figure R8. Kinetic profile of ITO/SnO₂ at 5,000 nm in the μ s region following excitation by the 1,400-nm laser (IRF: ca. 2 μ s). The red line shows the best fit.

Table R2. Decay components of transient absorption of ITO/SnO₂ at 5000 nm

	τ_1 (μ s)	τ_2 (μ s)	τ_3 (μ s)
ITO/SnO ₂ (S)	2.1	40	200
ITO/SnO ₂ (L)	3.5*	46	220

*This value affected by the light scattering caused by the large SnO₂ particles.

I would also request more electron microscopy images of the heterointerfaces formed and investigated, both to allow consistency between areas of the structures to be established, as well as some at higher magnification showing the interfacial area.

Response: Thank you for indication. According to your advice, we added the electron microscopy images (HAADF, HRTEM, and EDS mapping) of heterointerfaces. From the result of imaging of several ITO/SnO₂ crystal interfaces, random facets of ITO NC attach on SnO₂.

Action: We added the electron microscopy images of heterointerfaces in the supplementary Fig. 5 of revised manuscript.

Figure R9. **a**, HAADF-STEM image of ITO/SnO₂ heterointerface. **b**, HAADF-STEM-EDS elemental mapping images of ITO/SnO₂ heterointerface. **c**, HRTEM image of ITO/SnO₂ heterointerface

We hope that the revised version of our manuscript is now suitable for the publication in *Nature Communications*.

Sincerely yours,

Masanori Sakamoto
Prof. Dr.
Institute for Chemical Research,
Kyoto University,
Gokasho, Uji, Kyoto 611-0011, Japan
E-mail: sakamoto@scl.kyoto-u.ac.jp

Reviewers' comments:

Reviewer #1 (Remarks to the Author):

The author changed the MS carefully and they consider all of my comments and question. Therefore I recommend this MS for publication in nature communications.

Reviewer #2 (Remarks to the Author):

The reviewers addressed the comments from the previous round or revision satisfactory; however, I cannot recommend it for publication in Nature Communications. It is rather hard to connect the context of the abstract, introduction and concluding remarks. The efficient charge transfer with 37% percent is claimed for ITO/SnO₂; however, probably the wrong set of experiments has been chosen to demonstrate the advantage of this interface. Almost negligible photocurrents were detected for such system. Also I was surprised to find out that this study totally ignores the numerous relevant studies on conductive and transparent nanocrystals from Delia Milliron' group. Also, another important publication that discussed the aspects of interface design for efficient charge separation was not cited (Nature Photonics 2014, 8,95). The scholarly of presentation is not at the level of Nature' group journals. Minor comments are the following:

1. Figure 5(SI) - the EDS mapping data are, probably, not the most representative. Thus Figure 3(SI) presenting the size histograms indicates the sized of 11 and 25 nm for ITO and SnO₂, respectively. However, Figure 5(SI) shows features that are not in agreement with the data shown in Figure 3(SI).
2. Figure 1 - the schematic depiction of the interface is giant, while the optical image, as well as TEM image of ITO NPs are barely noticed.

Reviewer #3 (Remarks to the Author):

The time the authors have addressed the queries made by the reviewers is noted and appreciated. However, the description/quantification of the kinetics following photoexcitation within the manuscript still requires major revision prior to publication of the work.

The quantification of the kinetics following photoexcitation remains an issue. For the fs regime, the decay constants for each of the individual components are not reported. Whilst a Table showing one decay constant has been added to the SI for the decay, in the text referring to this table two decays occurring over different timescales are described (Page 3, lines 97 – 99, “Upon excitation of ITO or SiO₂, transient absorptions (ΔOD) appear within the time resolution of ~ 200 fs and decay with fast and slow components within 1 to 2 ps.”), along with a “rise”. It is also noted in the text that the decay occurs over 1 to 2 ps, however no rate constants/lifetimes with this time constant are reported. In the subsequent paragraph “Conversely, the ITO/SnO₂ and TiO₂ heterointerfaces also showed a distinctly strong rise within the time resolution, decaying within 1–2 ps, and a subsequent and much-longer lifetime component, with the fast and much-slower components being assignable to the above-mentioned LSPR-induced ultra-fast events and the FCA of SnO₂(or TiO₂), respectively.”, the lifetimes of the processes referred to by the authors are also not reported. It is unclear to me from the text exactly how many processes are being referred to. I would like to re-iterate that all decay constants, in all time regimes, should be accurately reported along with their errors. Also, the IRF for the fs measurements should be shown on the graphs and its width measured accurately (not approximately), and that for the us system should be accurately measured, not approximated. This is necessary to confirm their instrument time resolution and is particularly important for the fs regime. Similar issues are apparent in the discussion of the us kinetics.

It is unclear what is meant by the phrase/assignment “FCA affected by IRF” in the discussion of the us decay kinetics of their system. This should be clarified.

Do the authors mean “measured” where they have written “estimated”? Or are these values indeed estimated?

Answers to Reviewers' comments:

Reviewer #1 (Remarks to the Author):

The author changed the MS carefully and they consider all of my comments and question. Therefore, I recommend this MS for publication in nature communications.

Response: We appreciate your recommendation to *Nature Communication*.

Reviewer #2 (Remarks to the Author):

The reviewers addressed the comments from the previous round or revision satisfactory; however, I cannot recommend it for publication in Nature Communications.

It is rather hard to connect the context of the abstract, introduction and concluding remarks.

Response: We are sorry, but we cannot follow your indication. It would be appreciated if you indicate how hard to connect the context of the abstract, introduction and concluding remarks.

The efficient charge transfer with 37% percent is claimed for ITO/SnO₂; however, probably the wrong set of experiments has been chosen to demonstrate the advantage of this interface. Almost negligible photocurrents were detected for such system.

Response:

We should point out that the quantum yield of 33% is not a surprisingly high value as the plasmonic hot electron transfer systems. For example, quantum yield of ca. 40% was reported in Au/TiO₂ system by A. Furube (*J. Am. Chem. Soc.* 2007, 129, 14852-14853.), who is one of the authors of this work. This work is referred in a review article by C. Clavero (*Nat. Photon.* 2014, 8, 95), which you mentioned as an important review. Recently, Ratchford et al. reported the quantum yield of 25 to 45% in the Au/TiO₂ system (*Nano Lett.* 2017, 17, 6047.).

As for “the wrong set of experiments”, we clearly showed all the experimental set in the manuscript or supplementary information. If you point out what is “wrong” in detail, we are ready to respond your indication. Because we employed the similar set up to the previous works, which have been published in international journals (for example, *J. Am. Chem. Soc.* 2007, 129, 14852.), we strongly believe our experimental set is not “wrong” as you mentioned.

The structure of photoelectrode for photocurrent measurements is based on the structure of Au/TiO₂, which is a typical system for plasmonic photo-electro conversion system (for example, Shi *et al.*, *J. Phys. Chem. C* 2013, 117, 2494.). At the interface of Au/TiO₂, the formation of Schottky junction enables the plasmonic hot electron injection. In the present system, the ITO/SnO₂ and ITO/TiO₂ interfaces also form the Schottky junctions (Pfeifer *et al.*, *J. Phys. Chem. Lett.* 2013, **4**, 4182.). Therefore, our photoelectrode follows the conventional system for plasmonic photo-electricity conversion.

About the photo-response of ITO/SnO₂ electrode, we clearly observed the rise and decay of photocurrent following the on/off of light irradiation. In addition, the photo-response is repeatable. It is obvious that the IR light induced photocurrent is not negligible but significant. We should point out that the low photocurrent by LSPR induced charge separation is common in this field of research (For example, Shi *et al.*, *J. Phys. Chem. C* 2013, 117, 2494.). Generally, the low photocurrent is derived from the loss by the charge recombination or trapping. Furthermore, we need to mention that the investigation of technique to enhance the photocurrent is not the aim of present work.

Again, if you can provide us with the details of your indication "probably the wrong set of experiments has been chosen", we are ready to answer your question.

Also I was surprised to find out that this study totally ignores the numerous! relevant studies on conductive and transparent nanocrystals from Delia Milliron' group. Also, another important publication that discussed the aspects of interface design for efficient charge separation was not cited (Nature Photonics 2014, 8,95).

Response: Because we consider that the IR-induced hot electron transfer form transparent NCs is the most important discovery in the present work, we carefully selected the reference papers related to the LSPR-induced hot electron transfer. We agree with the importance of researches concerning optical properties of transparent NCs by Delia Milliron' group even if these research subjects are not hot electron transfer. We also agree with the importance of a general review by C. Clavero. Therefore, we added following references (7,8 and 24) in the revised version of manuscript.

Action: We added the references (7,8 and 24) in the revised version of manuscript.

The scholarly of presentation is not at the level of Nature' group journals.

Response: We are sorry, but we cannot follow your indication. It would be appreciated if you indicate how “the scholarly of presentation is not at the level of Nature' group journals”. We would like to mention that the manuscript has been carefully reviewed by an experienced person whose first language is English and who specializes in editing papers written by scientists whose native language is not English.

Minor comments are the following:

1. Figure 5(SI) - the EDS mapping data are, probably, not the most representative. Thus Figure 3(SI) presenting the size histograms indicates the sized of 11 and 25 nm for ITO and SnO₂, respectively. However, Figure 5(SI) shows features that are not in agreement with the data shown in Figure 3(SI).

Response: Thank you for your indication. For the investigation of heterointerfaces, we chose SnO₂(L) (US Research Nanomaterials, Inc., nominal diameter is 450 nm). The size distribution of SnO₂(L) was much larger than SnO₂(S) and agreed with supplementary Figure 3, as shown in Supplementary Figure 10 b.

Action: We changed the word SnO₂ to SnO₂(L) in the caption of Supplementary Figure 5 for clarity.

2. Figure 1 - the schematic depiction of the interface is giant, while the optical image, as well as TEM image of ITO NPs are barely noticed.

Response: Thank you for your indication.

Action: We revised Figure 1 according to your indication. We removed the TEM image of ITO NCs because the image is shown in Supplementary Figure 2. The optical image was enlarged as you indicated.

Reviewer #3 (Remarks to the Author):

The time the authors have addressed the queries made by the reviewers is noted and appreciated. However, the description/quantification of the kinetics following photoexcitation within the manuscript still requires major revision prior to publication of the work.

Response: We appreciate your comments and advices. We have conducted additional experiments to address all the comments. We hope that the explanation and revision of our work are clear enough.

The quantification of the kinetics following photoexcitation remains an issue.

For the fs regime, the decay constants for each of the individual components are not reported. Whilst a Table showing one decay constant has been added to the SI for the decay, in the text referring to this table two decays occurring over different timescales are described (Page 3, lines 97 – 99, “Upon excitation of ITO or SiO₂, transient absorptions (ΔOD) appear within the time resolution of ~ 200 fs and decay with fast and slow components within 1 to 2 ps.”), along with a “rise”. It is also noted in the text that the decay occurs over 1 to 2 ps, however no rate constants/lifetimes with this time constant are reported. In the subsequent paragraph “Conversely, the ITO/SnO₂ and TiO₂ heterointerfaces also showed a distinctly strong rise within the time resolution, decaying within 1–2 ps, and a subsequent and much-longer lifetime component, with the fast and much-slower components being assignable to the above-mentioned LSPR-induced ultra-fast events and the FCA of SnO₂(or TiO₂), respectively.”, the lifetimes of the processes referred to by the authors are also not reported.

Response: Thank you very much for your advices. We added the values indicated by you in the main text of revised manuscript.

Action: We added values of decay component in the line 99, and line 108 and rewrite the discussion for clarify. In addition, according to your advice, we added Tables 1 and 2, which show the decay constants in the ps or μs regime in the supplementary of revised version of manuscript.

It is unclear to me from the text exactly how many processes are being referred to. I would like to re-iterate that all decay constants, in all time regimes, should be accurately reported along with their errors.

Response: Thank you very much for your advices. Since the decay processes of ITO/SnO₂ and ITO/TiO₂ range from ps to μs region, we used the different instruments for TAS measurements for ps and μs regions.

We show a brief illustration about the temporal sequence of hot electron injection following the excitation of ITO NC in Figure R1. We can observe 2 processes, i.e., the carrier relaxation following the excitation of ITO (i.e., electron-electron scattering and electron-phonon coupling) and charge

recombination following the hot electron injection. In the present system, we cannot observe the processes faster than the time-resolution of system, such as hot electron injection and Landau damping. The decay components are shown in tables R1 and 2.

We did not underestimate the errors in kinetics, and only showed errors which can affect the measured values within the significant figures. Therefore, we did not show errors for the τ values in μ s region.

Figure R1. Brief illustration about the temporal sequence of hot electron injection following the excitation of ITO NC attached on SnO₂.

Table 1 | Decay (τ_1 and τ_2) components of LSPR in the ps region.

	τ_1 (ps)	τ_2 (ps)
ITO/SiO ₂	0.37±0.01	NA
ITO/SnO ₂	0.21±0.01	5.3±0.7
ITO/TiO ₂	0.14±0.01	4.5±0.7
ITO	0.17±0.01	NA

Table 2 | Decay components of transient absorption at 5000 nm of ITO/SnO₂ with different sizes in the μs region.

	τ_1 (μs)	τ_2 (μs)	τ_3 (μs)
ITO/SnO ₂ (S)	1.9	32	159
ITO/SnO ₂ (L)	2.9*	31	176

* The value affected by the light scattering of the large SnO₂ particles.

Action: According to your advice, we added Tables, which show the decay constants in the ps or μs regime in the revised version of supplementary.

Also, the IRF for the fs measurements should be shown on the graphs and its width measured accurately (not approximately), and that for the μs system should be accurately measured, not approximated. This is necessary to confirm their instrument time resolution and is particularly important for the fs regime. Similar issues are apparent in the discussion of the μs kinetics.

Response: We estimated the IRF value of fs-LFP system according to the previous work (*J. Phys. Chem. C* 2009, 113, 6454–6462). Figure R2 shows time-resolved transient absorption of Si measured by our instrument (a) and the deviation (b). The FWHM value of IRF was estimated to be 285 fs by the fitting of the curve by using the Gauss function (red line).

For the IRF value of μs -LFP system, we estimated the time resolution of the system from the scattering of laser pulse measured by the system as shown in Figure R3. The FWHM value of IRF was estimated to be 0.485 μs by the fitting of the curve by using the Gauss function (red line).

Figure R2. **a** transient absorption signal of Si wafer measured by the present system. **b** the derivation of the transient absorption signal. Red line is a best fit. The FWHM of IRF value was estimated to be 285 fs by the fitting of the curve by using the Gauss function (red line).

Figure R3. Estimation of instrument response function of μ s-transient absorption measurement system. The scattering of laser pulse measured by the present system. Red line is a best fit. The FWHM of IRF value was estimated to be 0.485 μ s by the fitting of the curve by using the Gauss function (red line).

Action: We added estimation of the accurate the FWHM value of IRF of fs- and μ s-LFP system in the supplementary Figure 6 and 12 respectively, in the revised version of manuscript. The IRF value in the previous version of manuscript was corrected by the accurate value. In addition, we revised all discussion related to the IRF values following their estimation. The all decay constants were estimated again by using the curve fitting taking obtained IRF value in consideration.

It is unclear what is meant by the phrase/assignment “FCA affected by IRF” in the discussion of the us decay kinetics of their system. This should be clarified.

Response: The sentence means the signals decay within the time resolution of our instrument. As you pointed out, the “FCA affected by IRF” is not correct.

Action: We rewrite the description including the phrase of “FCA affected by IRF” in the revised version of manuscript.

Do the authors mean “measured” where they have written “estimated”? Or are these values indeed estimated?

Response: We use “estimated” in the sense of “measured”, “calculated” and “determined”.

Action: We revised the word “estimated” suitably in the revised version of manuscript.

We hope that the revised version of our manuscript is now suitable for the publication in *Nature Communications*.

Sincerely yours,

Masanori Sakamoto

Prof. Dr.

Institute for Chemical Research,

Kyoto University,

Gokasho, Uji, Kyoto 611-0011, Japan

E-mail: sakamoto@scl.kyoto-u.ac.jp

REVIEWERS' COMMENTS:

Reviewer #2 (Remarks to the Author):

The authors obviously improved the manuscript. The quality of the figures is MUCH better now. The discussion of the results is easier to understand. It is still confusing to understand what the novelty is. If it is the transport of the LSPR electrons from one "transparent" semiconductor to another, the authors indeed demonstrated the proof of principle of this phenomenon. However, the photoresponse is really small (Figure 4) and "leaky". If the "electron-injection efficiency at ITO/SnO₂ interface is 33%", why the photocurrent is so low? I am not surprised by the high quantum yield of 33%. I just understood that 33% was used to describe the electron-injection efficiency at ITO/SnO₂ interface. This is very confusing. If these data are not important for the content of this study as it is claimed by the authors, why they were included in the main text?

The authors claim that "The selective-excitation of LSPR of ITO causes hot-electron injection with significantly high efficiency (33%) and extraordinarily long-lived charge separation (c.a. 2-200 μ s) thanks to fine control of the heterointerface". However, I do not see any discussion on the control of the interface besides the discussion of the different interface (ITO/SnO₂ and ITO/TiO₂, etc).

In the response, the authors claimed that since the focus of the manuscript is on the LSPR they selectively did not cite papers from Milliron's group. However, this group published a few nice papers on LSPR effects in doped SnO₂.

Reviewer #3 (Remarks to the Author):

The time the authors have addressed the queries made by the reviewers is noted and appreciated. The description/quantification of the kinetics following photoexcitation within the manuscript is significantly improved. The authors should 1) include full experimental detail of the instruments the kinetic experiments were carried out using, including pulse energy, and 2) revise the errors quoted for the ps (and microsecond) measurements prior to publication, as an error of 0.01 ps for lifetimes comparable with the IRF seem very low. Perhaps the authors are quoting the error from the refinement of the fit, rather than their experimental error. These two things should be addressed prior to publication.

Answers to Reviewers' comments:

Reviewer #2 (Remarks to the Author):

Thank you very much for your advices, which improved our paper significantly. We have addressed all the comments.

The authors obviously improved the manuscript. The quality of the figures is MUCH better now. The discussion of the results is easier to understand. It is still confusing to understand what the novelty is. If it is the transport of the LSPR electrons from one "transparent" semiconductor to another, the authors indeed demonstrated the proof of principle of this phenomenon.

Response: As you pointed out, the novelty is the transport of the LSPR-induced electrons from one transparent semiconductor (ITO) to another (SnO_2), which was demonstrated in the manuscript. We believe that our demonstration is a strong proof of principle of this mechanism.

However, the photoresponse is really small (Figure 4) and "leaky". If the "electron-injection efficiency at ITO/ SnO_2 interface is 33%", why the photocurrent is so low? I am not surprised by the high quantum yield of 33%. I just understood that 33% was used to describe the electron-injection efficiency at ITO/ SnO_2 interface. This is very confusing. If these data are not important for the content of this study as it is claimed by the authors, why they were included in the main text?

Response: As we mentioned in the previous manuscript, the carrier trapping and charge recombination following the hot electron injection is a reason for the low IPCE of photoelectrode. The "leak" does not concern the reduction of IPCE in the present system. This low IPCE is a common problem in the LSPR-induced energy conversion systems. Because the time-resolved transient absorption measurement can investigate the ultrafast charge separation, we can estimate the electron injection efficiency without the effect of the charge recombination. On the other hand, the photoresponse from the photoelectrode contains all events including the charge separation, recombination and trapping. Therefore, the photoresponse is small because the value is affected by the carrier trapping and charge recombination.

These data are so important in this paper because they clarified intrinsic hot carrier injection dynamics. We believe that the understanding of each step is requisite for understanding the whole picture of our system.

The authors claim that "The selective-excitation of LSPR of ITO causes hot-electron injection with significantly high efficiency (33%) and extraordinarily long-lived charge separation (c.a. 2-200 μs) thanks

to fine control of the heterointerface". However, I do not see any discussion on the control of the interface besides the discussion of the different interface (ITO/SnO₂ and ITO/TiO₂, etc).

Response: We made several interfaces between ITO and metal oxide (SnO₂, TiO₂, and SiO₂) to find the best combination. In addition, we established the effective way to fabricate the heterointerfacial contact between plasmonic ITO and metal oxides. We would like to claim that these processes are fine control of heterointerfaces.

In the response, the authors claimed that since the focus of the manuscript is on the LSPR they selectively did not cite papers from Milliron's group. However, this group published a few nice papers on LSPR effects in doped SnO₂.

Response: Thank you for your indication. We already referred the excellent works from Milliron's group related to heavily doped semiconductor according to your indication. Please find references 7 and 8 in the previous version of manuscript.

Reviewer #3 (Remarks to the Author):

Thank you very much for your advices, which improved our paper significantly. We have addressed all the comments.

The time the authors have addressed the queries made by the reviewers is noted and appreciated. The description/quantification of the kinetics following photoexcitation within the manuscript is significantly improved. The authors should 1) include full experimental detail of the instruments the kinetic experiments were carried out using, including pulse energy, and 2) revise the errors quoted for the ps (and microsecond) measurements prior to publication, as an error of 0.01 ps for lifetimes comparable with the IRF seem very low. Perhaps the authors are quoting the error from the refinement of the fit, rather than their experimental error. These two things should be addressed prior to publication.

Response: We appreciate your suggestive comments on the experimental details and uncertainty on the approximated lifetimes. In this regard, we modified the descriptions on the experimental method section, as follows.

“Microsecond (μ s) time-resolved IR-absorption measurements were conducted using custom-built spectrometers, as described in our previous papers. ITO/metal oxide samples were photoexcited by using a 1400 nm laser pulses (energy: 2.7 mJ pulse⁻¹, duration: 6 ns, repetition rate: 1 Hz) originating from a Nd: YAG laser (Continuum Surelite II) equipped with an optical parametric oscillator (OPO) system to generate the desired pump wavelength. The IR light emitted from the MoSi₂ coil was used as the probe light in the mid-IR region (7000 – 1000 cm⁻¹). The transmitted IR light from the ITO/metal oxide samples fixed on the CaF₂ plate was then introduced into the grating spectrometer

and the monochromated light from the spectrometer was detected by an MCT detector (Kolmar), and then the output electric signal was amplified using an AC-coupled amplifier (Standford Research System SR560, 1 MHz). The time resolution of the spectrometers was limited to 1 - 2 μ s by the bandwidth of the amplifier. The instrument response function (IRF) was evaluated by measuring the scattered laser pulses detected by the MCT. The FWHM of IRF value was estimated to be 0.485 μ s as indicated in Figure S12.

In the femtosecond-to-picosecond region, the ultrafast kinetic measurements were performed using on Ti:sapphire laser system (Spectra Physics, Solstice and TOPAS Prime, duration: 90 fs, repetition rate: 1 kHz) equipped with an OPO to generate the pump and probe wavelengths. The ITO/metal oxide samples were photoexcited using 1,700 nm (energy: 6 μ J pulse⁻¹). The probe light was focused on the sample and the transmitted IR light during irradiation condition entered the spectrometer equipped with gratings. The monochromated light was then detected by MCT detector. The time resolution of the spectrometer was \sim 90 fs, which limited by the temporal width of the laser pulse. The FWHM of IRF value was estimated to be 285 \pm 40 fs (refer to Figure S6b). For the measurement of the kinetic profile shown in Supplementary Fig. 14, a femtosecond Ti: sapphire laser system (Spectra Physics; Hurricane & TOPAS; wavelength: 800 nm; pulse duration: 150 fs; repetition rate: 1 kHz) was used. The IRF of the system is 210 fs. The 1,700-nm pulse from one OPA was used as a pump light. For the probe light, a 3,440-nm pulse generated from the other OPA with a difference-frequency-generation crystal was used. The intensity of the probe light transmitted from the sample was detected using an MCT photodetector (KMPV11-1-J1, Kolmar technology).”

We consider that the uncertainty (error) in ps experiment is mainly derived the error of FWHM of IRF and uncertainty of time origin. Therefore, we revised the errors taking the uncertainties of FWHM of IRF (285 \pm 40 fs) and time origin (\pm 0.05 ps) into account. We revised the errors for microsecond measurements from the experimental error, as you suggested.

Action: 1) We revised kinetic experiment including full experimental detail of the instruments. 2) We revised error taking the uncertainties of FWHM of IRF (285 \pm 40 fs) and time origin (\pm 0.05 ps) into account. We revise the errors for microsecond measurements from the experimental error. Please see Tables shown in below.

Table 1 | Decay (τ_1 and τ_2) components of LSPR in the ps region.

	τ_1 (ps)	τ_2 (ps)
ITO/SiO ₂	0.37 \pm 0.11	NA
ITO/SnO ₂	0.21 \pm 0.10	5.3 \pm 0.7
ITO/TiO ₂	0.14 \pm 0.09	4.5 \pm 0.7

ITO	0.17±0.08	NA
-----	-----------	----

Table 3 | Decay components of transient absorption at 5000 nm of ITO/SnO₂ with different sizes.

	τ_1 (μ s)	τ_2 (μ s)	τ_3 (μ s)
ITO/SnO ₂ (S)	2.0±0.1	33±1	160±1
ITO/SnO ₂ (L)	3.2±0.4*	31±2.8	179±4.2

* The value affected by the light scattering caused by the large SnO₂ particles.

We hope that the revised version of our manuscript is now suitable for the publication in *Nature Communications*.

Sincerely yours,

Masanori Sakamoto

Prof. Dr.

Institute for Chemical Research,

Kyoto University,

Gokasho, Uji, Kyoto 611-0011, Japan

E-mail: sakamoto@scl.kyoto-u.ac.jp